# The effect of body mass index reduction on intraocular pressure in a large prospective cohort of apparently healthy individuals in Israel

**Dan Coster[1,2], Ariel Rafie[2], Noam Savion-Gaiger**[3]*****, **Rony Rachmiel[2,3], Shimon Kurtz[2,3], Shlomo Berliner[2,4], Itzhak Shapira[2,4], David Zeltser[2,4], Ori Rogowski[2,4], Shani Shenhar-Tsarfaty[2,4], Michael Waisbourd[2,3]**

**1** Blavatnik School of Computer Science, Tel-Aviv University, Tel-Aviv, Israel, **2** Sackler Faculty of Medicine, Tel Aviv University, Tel-Aviv, Israel, **3** Division of Ophthalmology, Tel Aviv Medical Center, Affiliated to the Sackler Faculty of Medicine, Tel Aviv University, Tel-Aviv, Israel, **4** Division of Internal Medicine, Tel-Aviv Sourasky Medical Center, Affiliated to the Sackler Faculty of Medicine, Tel-Aviv University, Tel-Aviv, Israel

\* noam6675@gmail.com

**Data Availability Statement:** The underlying minimal data for this study cannot be shared publicly because of restrictions imposed by

## Abstract

### Purpose

To investigate the effect of change in body mass index (BMI) on intraocular pressure (IOP) in a large cohort of apparently healthy volunteers who underwent an annual comprehensive screening examinations.

### Methods

This study included individuals who were enrolled in the Tel Aviv Medical Center Inflammation Survey (TAMCIS) and had IOP and BMI measurements at their baseline and follow up visits. Relationships between BMI and IOP and the effect of change in BMI on IOP were investigated.

### Results

A total of 7,782 individuals had at least one IOP measurement at their baseline visit, and 2,985 individuals had $\geq$2 visits recorded. The mean (SD) IOP (right eye) was 14.6 (2.5) mm Hg and mean (SD) BMI was 26.4 (4.1) kg/m$^2$. IOP positively correlated with BMI levels (r = 0.16, p<0.0001). For individuals with morbid obesity (BMI$\geq$35 kg/m$^2$) and $\geq$2 visits, a change in BMI between the baseline and first follow-up visits correlated positively with a change in the IOP (r = 0.23, p = 0.029). Subgroup analysis of subjects who had a reduction of at least 2 BMI units showed a stronger positive correlation between change in BMI and change in IOP (r = 0.29, p<0.0001). For this subgroup, a reduction of 2.86 kg/m$^2$ of BMI was associated with a reduction of 1 mm Hg in IOP.

hospital privacy policies. Data are available from Aneta David, a member of Helsinki committee, via email (anetad@tlvmc.gov.il), for researchers who meet the criteria for access to confidential data.

**Funding:** The authors received no specific funding for this work.

**Competing interests:** The authors have declared that no competing interests exist.

## Conclusions

BMI loss correlated with reduction in IOP, and this correlation was more pronounced among morbidly obese individuals.

## Introduction

Glaucoma, an optic neuropathy characterized by progressive loss of retinal ganglion cells and their axons, is the leading cause of irreversible blindness worldwide [1, 2]. While there are several known risk factors for developing glaucoma, elevated intraocular pressure (IOP) is the most important and only modifiable risk factor for glaucoma [3–5]. Glaucomatous damage caused by increased IOP is believed to be mediated by mechanical compression and vascular compromise, according to one of the major theories for disease pathogenesis [6, 7]. It is therefore important to examine factors that may relate to an increase in IOP.

Obesity is considered a modern epidemic. According to the World Health Organization, more than 1.9 billion adults are overweight, and overall world obesity tripled since 1975 [8]. Moreover, since 1980 there has been a drastic increase in rates of morbid obesity in the US. [9, 10].

Several studies found positive correlation between metabolic risk factors, such as increased systolic and/or diastolic blood pressure, waist circumference, BMI and insulin resistance and IOP [11–15]. In a meta-analysis of 12 studies carried out by the European Eye Epidemiology (E3) Consortium, Khawaja et al. examined cross-sectional associations with IOP in 43,500 European adults, and found strong linear relationship between BMI and IOP across the entire range of BMI [16]. In a prospective study, Geloneck et al. measured IOP in 125 subjects across varying BMI ranges and found that higher BMI was correlated with higher IOP [17].

While numerous studies have shown an association between BMI and IOP, the effect of weight loss or weight gain on IOP has not been studied extensively. Mori et al. published a longitudinal study with a large cohort which demonstrated a significant positive trend between slope in IOP (defined as mmHg/years of age) and slope of weight (defined as kg/years of age) [18]. Lam et al. followed a cohort of 25 morbidly obese patients scheduled for bariatric surgery and observed that weight loss was weakly associated with a lowering of IOP [19].

The purpose of our study was to investigate the effect of change in BMI on IOP levels in a large population-based screening program in Tel Aviv, Israel.

## Methods

### Study population

The Tel-Aviv Medical Center Inflammation Survey (TAMCIS) is an ongoing, prospective, population-based study carried out at the Division for Preventive Medicine of the Tel-Aviv Medical Center, Israel. Participants were enrolled from a large screening program–the Tel-Aviv Medical Center's Executive Health Program. As part of this program, candidates for a comprehensive medical assessment were requested to enroll in the study and sign an informed consent form. A full, comprehensive medical assessment was coordinated by a senior internal medicine specialist. A complete physical examination was performed by an internal medicine specialist. Comprehensive laboratory and ancillary diagnostic tests were performed, including blood chemistry and metabolic profile, complete blood count, blood inflammatory markers, urine tests, occult fecal blood test, prostate-specific antigen blood test (for men>40 years), cardiac stress test, spirometry, audiometry, chest X-ray, and for women also gynecological

examination, including PAP test, physical breast examination, mammography, and breast ultrasound).

A comprehensive ophthalmic evaluation included ocular history, visual acuity, slit lamp examination by an ophthalmologist, and IOP measurement (Tonopen XL, Reichert, Inc., Depew, NY) by an ophthalmic technician. If IOP was deemed elevated by the examining ophthalmologist, it was repeated by Goldmann Applanation Tonometry, and the result was recorded as the final IOP.

Body weight was measured using an electric scale accurate to 5 grams with a weight limit of 200 kg. Height was measured using a 2-meter stadiometer coupled to the scale. BMI was calculated according to the WHO guidelines, with weight in kilograms divided by height in meters squared (kg/m$^2$). BMI loss was merely observed and was not part of a standard weight loss protocol.

We included in our analysis all individuals who had at least one IOP and BMI measurements at their baseline visit. Patient who did not have both IOP and BMI measured at their baseline visit were excluded.

The study was approved by the Tel Aviv Medical Center institutional review board (IRB # 02–049) and written informed consent was obtained from all participants

### Statistical analysis

IOP of the right eye of all eligible individuals was used for the current analysis. To compare the distributions of risk factors on the IOP we used the Anova test.

$\Delta IOP$ was defined as the IOP difference between the baseline IOP and first follow-up visit IOP. The $\Delta$ parameter was calculated similarly for the BMI (i,e., $\Delta BMI$ was the BMI difference between the baseline and first follow-up visit).

The Pearson correlation index was used to assess the correlation between IOP or $\Delta IOP$ and risk factors and the $\Delta BMI$. Pearson correlation was used to estimate correlation coefficients and using $\alpha = 0.05$ and expected correlation coefficient of 0.15 the required sample size that is required to provide statistical power of 0.99 is 807 [20]. A p-value of the correlation was estimated using the student's t-test for correlation. Mean and standard deviation of IOP levels were calculated in five BMI categories: BMI<18.5, 18.5≤BMI<25, 25≤BMI<30, 30≤BMI<35, and BMI≥35 kg/m$^2$. A multivariable linear regression model included baseline parameters of gender, age, smoking status, and BMI category as independent variables, and the IOP level as the continuous dependent variable. Patients with records suggesting that they had an active inflammation were removed from the analysis.

The statistical analysis was performed using the Python programming language version 3.5.2 and the packages SciPy, NumPy, and scikit-learn.

### Results

A total of 7,782 individuals who were enrolled between January 1, 2001, to December 31, 2017, and had IOP and BMI measurements were included in our analysis (Fig 1). Demographic and clinical characteristics of the study cohort appear in Table 1. The population's mean age was 47.4 years and older age was associated with higher IOP levels. Table 2 compares IOP levels across different subgroups. Most participants in the study cohort were male (n = 5,443, 70.0%) and the male gender had significantly higher IOP levels. BMI categories were divided as follows: underweight (BMI<18.5kg/m$^2$), n = 62 (0.8%); healthy weight (18.5kg/m$^2$≤BMI<25kg/m$^2$), n = 3,077 (39.5%); overweight (25kg/m$^2$≤BMI<30kg/m$^2$), n = 3,351 (43.1%); obese (30kg/m$^2$≤BMI<35Kg/m$^2$), n = 1,017 (13.1%); morbidly obese (35Kg/m$^2$≤BMI), n = 275 (3.5%). The mean (SD, Range) IOP was 14.6 (2.5, 5–31) mmHg (right eye)

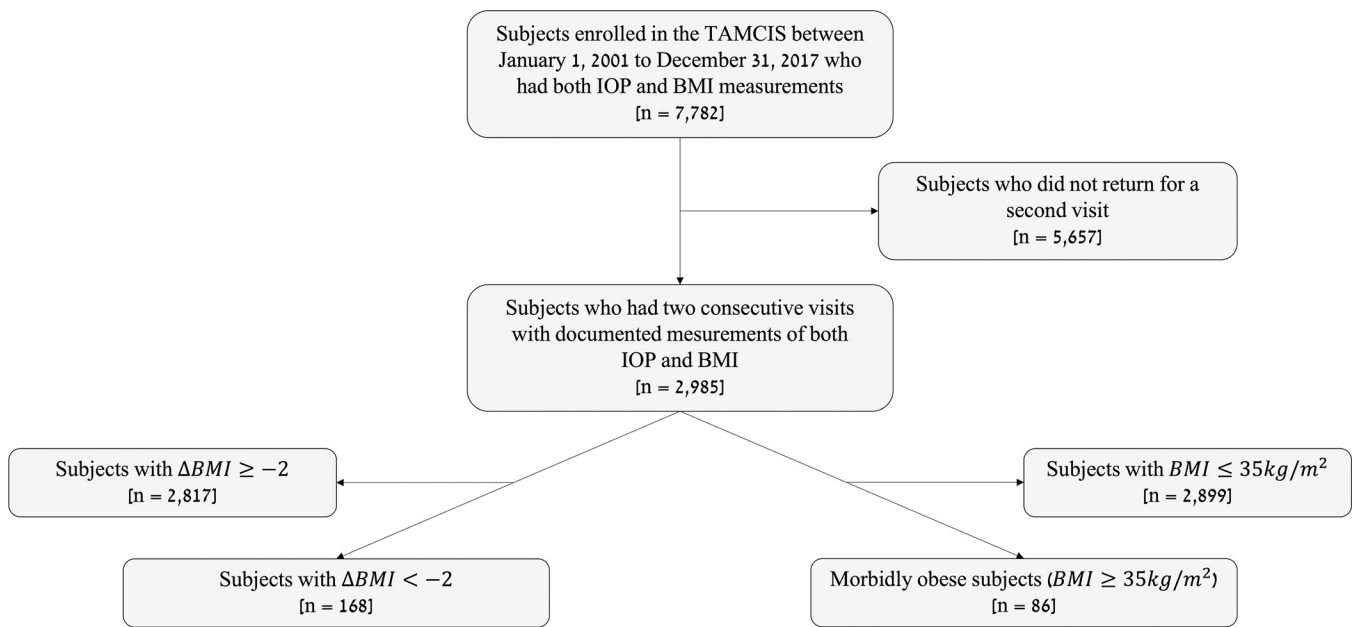

**Fig 1. Flow chart of the current study.** TAMCIS = Tel Aviv medical center inflammation survey, IOP = intraocular pressure, BMI = body mass index (kg/m$^2$).

and mean (SD) BMI was 26.4 (4.1) kg/m$^2$ in the general cohort. Higher BMI levels were associated with higher IOP levels. Smoking categories were divided into three categories: current smoker (11.1%), previous smoker (26.2%), and non-smoker (62.8%). History of smoking was associated with higher IOP levels.

Table 3 presents multivariate linear regression where the continuous dependent variable was the IOP level, and the independent variables were age, gender, BMI, and smoking status.

Table 1. Baseline characteristics of the study cohort (n = 7,782).

| Parameter | | Mean (SD) |
|---|---|---|
| Age, years | | 47.4 ± 10.7 |
| Vital signs | | |
| | Pulse, bpm | 69.6 ± 12.1 |
| | Diastolic BP, mm Hg | 77.3 ± 9.4 |
| | Systolic BP, mm Hg | 124.9 ± 14.8 |
| | Body temperature, ˚C | 36.4 ± 0.3 |
| Physical properties | | |
| | Weight, kg | 78.1 ± 14.8 |
| | Waist circumference, cm | 93.1 ± 11.6 |
| | Height, m | 1.72 ± 0.1 |
| Laboratory indices | | |
| | CRP mg/L | 2.6 ± 3.8 |
| | WBC 10$^9$/L | 6.7 ± 1.6 |
| | Hemoglobin A1C, % | 5.5 ± 0.5 |
| | Glucose, mg/dL | 85.7 ± 13.8 |
| | Hematocrit, % | 42.7 ± 3.5 |

bpm = beats per minute, BP = blood pressure, CRP = c-reactive protein, WBC = white blood cells.

Data are presented as means ± SD.

**Table 2. Comparison of intraocular pressure levels across different sub-groups (total cohort: n = 7,782).**

| | | N (%) | IOP (Mean ± SD) | p-value |
|---|---|---|---|---|
| Gender | | | | **< 0.00001** |
| | Male | 5,443 (70.0%) | 14.8 ± 2.5 | |
| | Female | 2,336 (30.0%) | 14.3 ± 2.4 | |
| Age, years | | | | **< 0.00001** |
| | 18–29 | 330 (4.2%) | 14.1 ± 2.5 | |
| | 30–39 | 1,685 (21.6%) | 14.3 ± 2.4 | |
| | 40–49 | 2,677 (34.4%) | 14.4 ± 2.5 | |
| | 50–59 | 1,954 (25.1%) | 14.9 ± 2.5 | |
| | $\geq$60 | 1,131 (14.5%) | 14.9 ± 2.5 | |
| BMI, kg/m$^2$ | | | | **< 0.00001** |
| | Underweight | 62 (0.8%) | 13.4 ± 2.5 | |
| | Normal weight | 3,077 (39.5%) | 14.2 ± 2.5 | |
| | Overweight | 3,351 (43.1%) | 14.8 ± 2.4 | |
| | Obese | 1,017 (13.1%) | 15.1 ± 2.5 | |
| | Morbidly obese | 275 (3.5%) | 15.5 ± 2.6 | |
| Smoking | | | | 0.0008 |
| | Currently smoker | 855 (11.1%) | 14.6 ± 2.4 | |
| | Previously smoking | 2,024 (26.2%) | 14.7 ± 2.5 | |
| | Non-smoker | 4,851 (62.8%) | 14.5 ± 2.5 | |

*Note*: Statistically significant p-values are in bold.

IOP = intraocular pressure, BMI = body mass index.

Underweight = BMI<18.5, normal weight = 18$\leq$BMI<25, overweight = 25$\leq$BMI<30

obese = 30$\leq$BMI<35, morbid obese = BMI$\geq$35.

In this multivariate analysis older age (OR 1.21 [1.15–1.28], p < 0.00001) and higher BMI (OR 1.08 [1.02–1.15], p = 0.005), were significantly associated with higher IOP levels whereas female gender (OR 0.85 [0.80–0.90], p < 0.00001) was significantly associated with lower IOP.

**Fig 2** illustrates the distribution of IOP across different BMI categories. We divide the cohort into 5 subgroups based on BMI measurements at the baseline visit. For most BMI categories, there was a significant increase in median and interquartile IOP levels. **Fig 3** shows a positive linear correlation between BMI measures at baseline visit and IOP (n = 7,782, r = 0.16, p<0.0001), and lack of correlation between IOP and height.

A subgroup analysis of individuals who had two visits or more was conducted (n = 2,985). The mean (SD) time difference between the first and second visit was 1.95 (0.9) years. The differences between the first and second visits for $\Delta BMI$ and $\Delta IOP$ were calculated with mean (SD) 0.05 (1.5) kg/m$^2$ and -0.29 (2.57) mm Hg respectively. The correlation between $\Delta BMI$ and $\Delta IOP$ among individuals with BMI$\geq$35 kg/m$^2$ at baseline is shown in **Fig 4A** (n = 86; r = 0.23, p = 0.029). Subgroup analysis of individuals who had $\Delta BMI$<−2 is presented in **Fig 4B**. In this group, a stronger correlation was found between $\Delta BMI$ and $\Delta IOP$ (r = 0.29, p<0.0001). For this subgroup, a reduction of 2.86 kg/m$^2$ of BMI was associated with a reduction of 1 mmHg in IOP.

## Discussion

In this large population-based cohort study (N = 7,782), we found that BMI positively correlated with IOP levels (r = 0.16, p<0.0001). We also showed that BMI reduction between

**Table 3. Multivariable analysis of factors associated with intraocular pressure (n = 7,782).**

| | | Beta coefficient (95% CI) | Odds Ratio | p-value |
|---|---|---|---|---|
| Sex (reference = men) | | | | |
| | Female | -0.16 | 0.85 | < **0.00001** |
| | | (-0.22: -0.1) | (0.80: 0.90) | |
| Age, years* | | 0.19 | 1.21 | < **0.00001** |
| | | (0.14: 0.25) | (1.15: 1.28) | |
| BMI, kg/m$^2$ (reference = overweight) | | | | |
| | Underweight | -0.09 | 0.91 | **0.002** |
| | | (-0.14: -0.03) | (0.87–0.97) | |
| | Normal weight | -0.21 | 0.81 | < **0.00001** |
| | | (-0.27: -0.15) | (0.17: 0.86) | |
| | Obese | 0.08 | 1.08 | **0.005** |
| | | (0.02: 0.14) | (1.02: 1.15) | |
| | Morbid obese | 0.13 | 1.13 | < **0.00001** |
| | | (0.08: 0.19) | (1.08: 1.21) | |
| Smoking (reference = non-smoker) | | | | |
| | Previous smoker | 0.04 | 1.04 | 0.21 |
| | | (-0.02: 0.09) | (0.98: 1.09) | |
| | Current smoker | 0.04 | 1.04 | 0.13 |
| | | (-0.01: 0.10) | (0.99: 1.11) | |

*Age was calculated as a continuous variable

*Note*: Reference group is the largest group. Statistically significant p- values are highlighted in bold (p<0.05).

IOP = intraocular pressure, CI = confidence interval, BMI = body mass index.

Underweight = BMI<18.5, normal weight = 18≤BMI<25, Overweight = 25≤BMI<30

Obese = 30≤BMI<35, Morbidly obese = BMI≥35.

Beta coefficient are the regression coefficients for standardized data.

baseline and follow-up visits correlated with a decrease in IOP among individuals with morbid obesity (defined as BMI≥35 kg/m2; r = 0.23, p = 0.029). Moreover, a more pronounced correlation (r = 0.29, p<0.0001) was found between Δ*BMI* and Δ*IOP* among subjects who lost over 2 kg/m$^2$ BMI units.

Our results agree with a number of cross-sectional studies that demonstrated a positive correlation between IOP and BMI from the USA [21, 22], Korea [23], Thiland [24], Israel [15], and China [25]. Other studies investigating the effect of BMI reduction on IOP levels reported conflicting results. While ElShazly et al. found no significant change in IOP following bariatric surgery [26], others did show a significant decrease in IOP following weight loss surgery [27, 28]. The effect of weight loss on IOP was also evident in several longitudinal population-based studies from Japan [18, 29] and China [25, 30].

In the current study, we found that for individuals who had a reduction of 2 BMI units or more, reduction of 2.86 kg/m$^2$ of BMI was associated with decrease of 1 mmHg in IOP. In comparison, Cohen et al reported that reduction of BMI of 10 kg/m$^2$ was associated with mean IOP reduction of 0.9- and 0.7- mm Hg for men and women, respectively [15]. Another study reported that 10% decrease in body weight induced 1.4 mm Hg reduction in IOP [19]. Theses results are consistent with the convention that greater weight loss is likely to achieve better health outcomes [31].

Several mechanisms have been proposed to explain how obesity affects IOP: (1) Excess intra-orbital fat tissue may cause episcleral venous compression and choroidal vascular

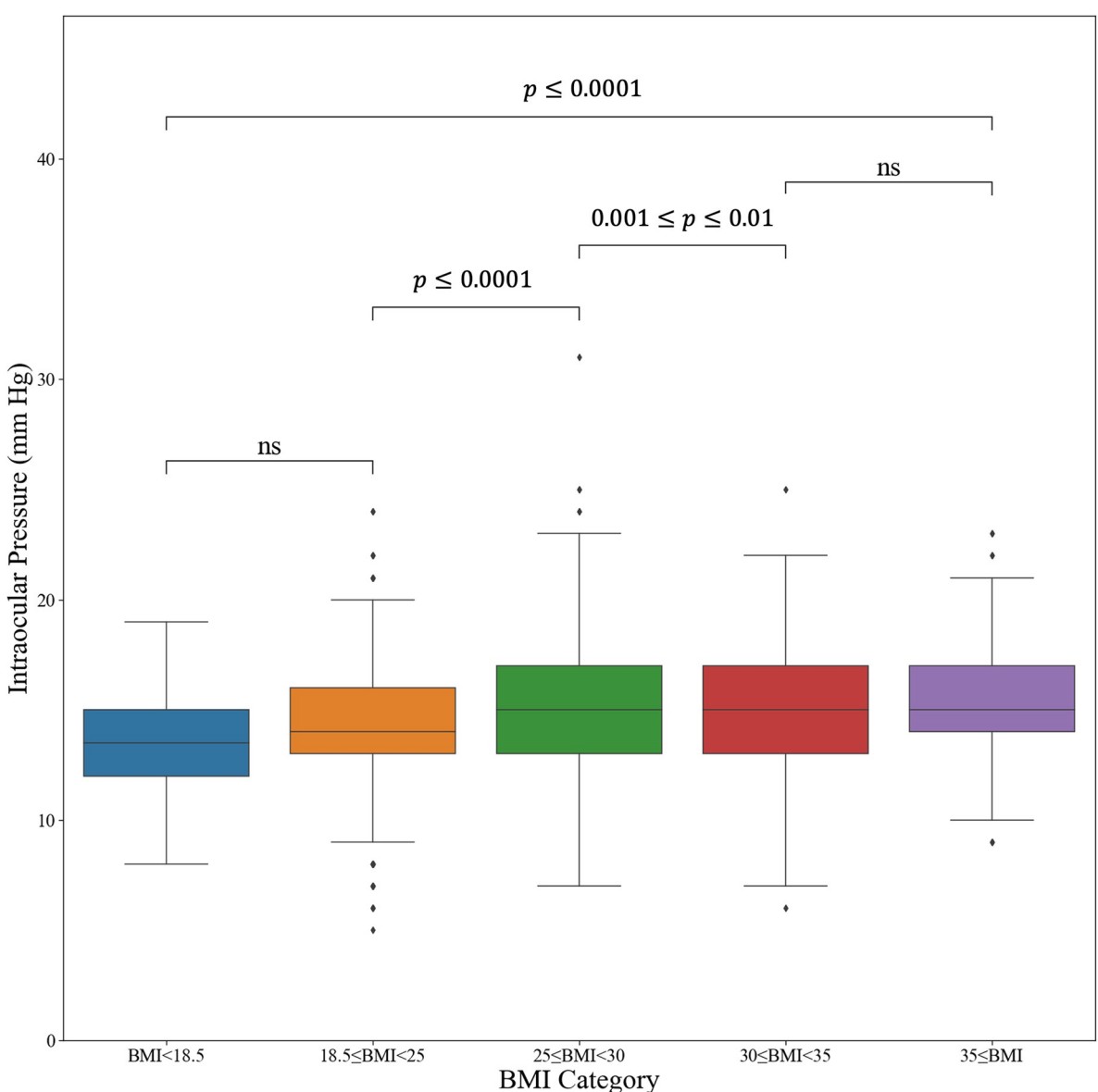

**Fig 2. Box plot comparing intraocular pressure levels across different body mass index categories.** IOP = intraocular pressure, BMI = body mass index, ns = non-significant.

congestion, subsequently leading to IOP increase [32] (2) Obesity is associated with higher blood viscosity, which decreases aqueous outflow [18] (3) Increased secretion of endogenous steroids induces an increase in IOP [23] (4) Chronic oxidative stress related to obesity contributes to malfunction of the outflow pathway by impairing the intracellular proteasome system of the cells [33–35] (5) Obesity is closely related to systemic conditions, including diabetes and hypertension, that are well recognized as risk factors for ocular hypertension [15, 18, 23, 36, 37] (6) Obesity is a cofactor for vascular dysregulation, such as local vasospasms and disturbed autoregulation of blood flow in the optic nerve head, choroid, and other ocular tissues, which contributes to increase in IOP [38]. (7) We also suggest that an increase in BMI results in accumulation of fat in the neck area and thereby reduces venous return, increases episcleral venous pressure, and thereby increases IOP.

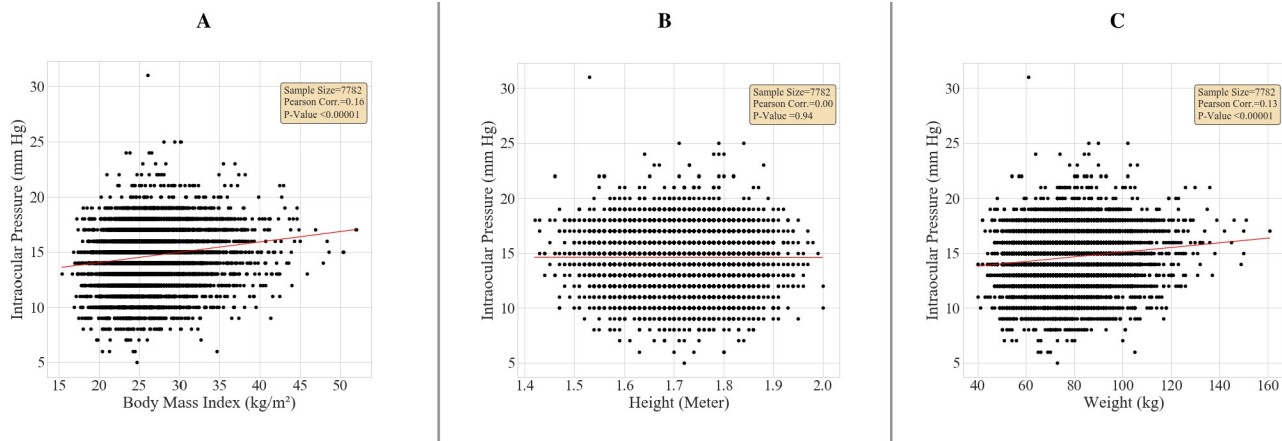

**Fig 3. Univariate regression analysis between intraocular pressure and obesity parameters (height, weight and body mass index) in the total cohort (n = 7,782). A**–Correlation between IOP and BMI. **B**–Correlation between IOP and height. **C**–Correlation between IOP and weight. IOP = intraocular pressure, BMI = body mass index.

Although our results were found in apparently healthy subjects, we can assume that BMI reduction will likely have a protective effect on patients with glaucoma, especially among morbidly obese individuals, since decrease of 1 mm Hg in IOP reduces the risk of glaucoma progression by 10% [39].

Our study also found a positive correlation between age and IOP, in agreement with other epidemiological studies [40–43].

Several studies evaluateed the association between smoking status and glaucoma, with contradicting results, and the direct impact of smoking on IOP reamins unclear [44–46]. In the current study, individuals who never smoked had slightly lower IOP compared to current or past smokers (14.5 mm Hg compared to 14.6 mm Hg and 14.8 mm Hg, respectively,

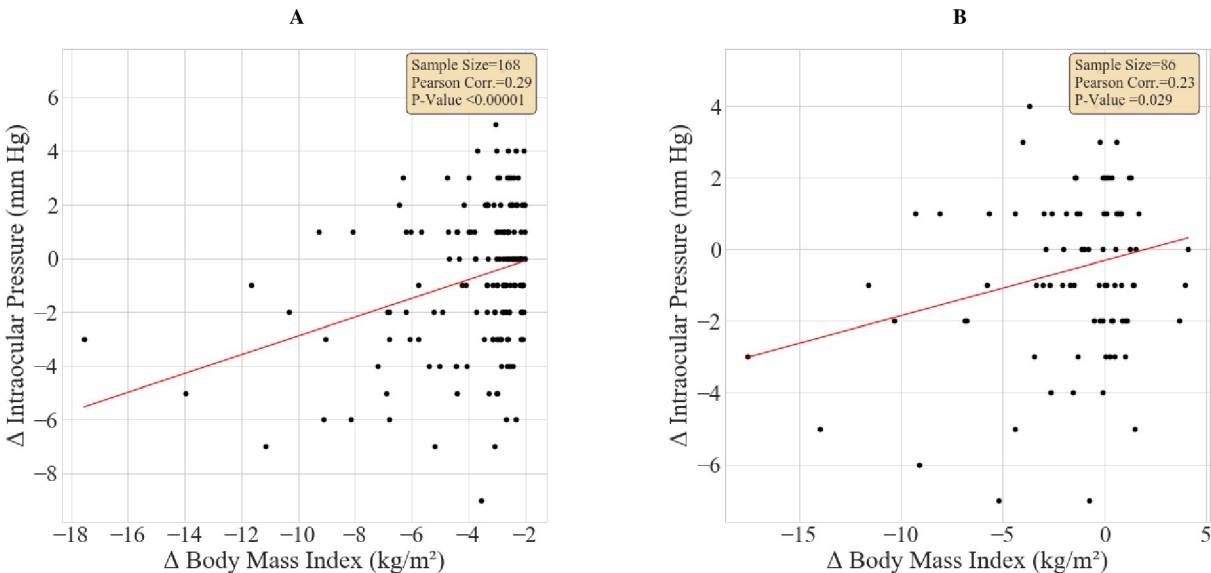

**Fig 4. Correlation between the differences in body mass index and intraocular pressure measurements from baseline to second visit. A**–Morbidly obese (n = 86). **B**–Subjects with $\Delta BMI < -2$ (n = 168). IOP = intraocular pressure, BMI = body mass index. Morbidly obese subjects = BMI$\geq$35kg/m$^2$.

p = 0.0008), although multiple variable analysis did not show significant differences between smoking status groups. This is in comparison to a large cohort study that reported higher IOP by 0.92 mmHg in current and past smokers compared to non-smokers [47].

Diabetes is a well established risk factor for developing glaucoma, and both prolonged duration of diabetes and higher fasting glucose levels are associated with increased risk for elevated IOP [48, 49]. Since diabetes is often associated with the metabolic syndrome (triad of diabetes melitus, obesity and elevated blood pressure,) it is likely that weight loss in morbid obese individuals will improve their blood glucose levels which may have additional positive impact on ocular health, including reduced risk for diabetic retinopathy and development of glaucoma.

Strengths of our study include the relatively large sample size, as well as the prospective, longitudinal assessment of two consecutive BMI and IOP measurements, which were done in only very few prior studies. We also used the gold standard Goldman applanation tonometery in addition to the Tonopen XL tonometer to accurately confirm IOP measurements >21 mm Hg. Our study also has several limitations. First, this is a single-center study, limiting the generalizability of our findings. Second, IOP and BMI were determined based on snapshot single measurements. The ideal alternative would have been a diurnal IOP curve, which was not feasible in such a large study. Third, the study enrolled apparently healthy individuals, and the direct impact of BMI loss on patients with glaucoma was not addressed. Forth, we included in our analysis individuals who had IOP measurements, regardless of the presence of other eye diseases including glauocma, which could have been a confounding factor.

In conclusion, this prospective, longitudinal study confirms the positive correlation between BMI and IOP and highlights the effect of BMI reduction on IOP. The strongest effect on IOP was noted in individuals who were morbidly obese (BMI>35kg/m$^2$). The implications on prevention of glaucoma in healthy individuals and possible protective effect in patients with glaucoma require further investigation.

## Author Contributions

**Conceptualization:** Michael Waisbourd.

**Formal analysis:** Dan Coster.

**Methodology:** Michael Waisbourd.

**Project administration:** Michael Waisbourd.

**Resources:** Michael Waisbourd.

**Supervision:** Michael Waisbourd.

**Visualization:** Michael Waisbourd.

**Writing – original draft:** Dan Coster, Ariel Rafie, Noam Savion-Gaiger, Michael Waisbourd.

**Writing – review & editing:** Rony Rachmiel, Shimon Kurtz, Shlomo Berliner, Itzhak Shapira, David Zeltser, Ori Rogowski, Shani Shenhar-Tsarfaty, Michael Waisbourd.

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
