## [Decision Letter · Decision Letter 0]

9 Mar 2023

PONE-D-23-02619The effect of body mass index reduction on intraocular pressure in a large prospective cohort of apparently healthy individuals in IsraelPLOS ONE

Dear Dr. Savion Gaiger,

Thank you for submitting your manuscript to PLOS ONE. After careful consideration, we feel that it has merit but does not fully meet PLOS ONE’s publication criteria as it currently stands. Therefore, we invite you to submit a revised version of the manuscript that addresses the points raised during the review process.

The authors have done a commendable job by recruiting so many participants, however, there is further scope of improvement in methodology, results and discussion as suggested below.

We look forward to receiving your revised manuscript.

Kind regards,

Natasha Gautam, MBBS, MS

Academic Editor

PLOS ONE

Journal Requirements:

Additional Editor Comments:

Though the sample size seems robust, it would be appreciated if the authors could comment on the power of study based on sample size and effect size.

Reviewers' comments:

Reviewer's Responses to Questions

**Comments to the Author**

1. Is the manuscript technically sound, and do the data support the conclusions?

Reviewer #1: Partly

Reviewer #2: Partly

2. Has the statistical analysis been performed appropriately and rigorously? 

Reviewer #1: Yes

Reviewer #2: Yes

3. Have the authors made all data underlying the findings in their manuscript fully available?

Reviewer #1: Yes

Reviewer #2: Yes

4. Is the manuscript presented in an intelligible fashion and written in standard English?

Reviewer #1: Yes

Reviewer #2: Yes

5. Review Comments to the Author

Reviewer #1: In the manuscript entitled “The effect of body mass index reduction on intraocular pressure in a large prospective cohort of apparently healthy individuals in Israel“ it’s commendable that a large number of subjects were included as well a long period of examination, but the real goal of the examination remains unclear; is it: can we prevent glaucoma by losing body weight? Or?

Although the authors report some interesting findings, the following issues need to be addressed:

1. The authors begin the introduction with the definition of glaucoma, but it is not clear in the method section whether the subjects have glaucoma or some other eye disease.

2. In Methods is stated that ophthalmic evaluation included ocular history, visual acuity and slit lamp; is it enough to confirm/rule out the existence of glaucoma or some other eye disease?

3. Methods: Which was inclusion/exclusion criteria for subjects who had IOP measured?

4. The authors should explain why the subjects had the following examinations: comprehensive laboratory and ancillary diagnostic tests, including blood chemistry and metabolic profile, complete blood count, blood inflammatory markers, urine tests, occult fecal blood test, prostate-specific antigen blood test (for men>40 years), cardiac stress test, spirometry, audiometry, chest X-ray, and for women also gynecological examination, including PAP test, physical breast examination, mammography, and breast ultrasound.

5. Methods: When was the second visit? What time period after the first visit?

6. Results: Which was baseline IOP range?

7. Have any ophthalmological diseases appeared during the follow-up period?

8. Why are laboratory indices indicated in Table 1?

9. What is clinical practice?

Reviewer #2: The authors present an interesting study on the association between body mass index and intraocular pressure and how weight loss leads to a decrease in intraocular pressure. The study covers a large number of patients, appropriate for a study of this type in which patients have to be subdivided into different subgroups according to body mass index. However, the authors comment that the study exclusively includes apparently healthy individuals, but the methodology described does not seem to support this statement sufficiently. Pachymetry or examinations of the optic disc are not performed, so it cannot be ruled out that certain individuals have glaucoma (normotensive, or that by making only an IOP measurement, possible ocular hypertension has not been detected due to IOP fluctuations , thin pachymetries etc). Furthermore, it is not clear whether individuals with IOPs above normal after two IOP measures were withdrawn from the study. It is commented that individuals who have weight loss have experienced a decrease in IOP but it is not detailed whether this weight loss has been part of a standardized protocol to which all obese patients have been subjected, to carry out the descriptive study or has been an individual weight loss according to the criteria of each patient. The discussion is a bit simple and poor, there are no comments on the glaucoma association with all the blood parameters that have been determined and explained in the results and neither with the height of the patients and the smoking habit; the bibliography is somewhat outdated; The possible association between the measured parameters and IOP, and its possible clinical relevance, need to be further explained. In the abstract, the conclusion part contains two sentences that are redundant since they say the same thing but with different words.

The beta coefficient of Table 3 is not explained in the table legend nor in the text; the meaning and interpretation of it should be addressed in order to better understand the results.

6. PLOS authors have the option to publish the peer review history of their article (what does this mean?). If published, this will include your full peer review and any attached files.

Reviewer #1: No

Reviewer #2: No

---

## [Author Response · Author response to Decision Letter 0]

21 Apr 2023

Dear Dr. Chenette, 

Thank you very much for considering our manuscript entitled: “The effect of body mass index reduction on intraocular pressure in a large prospective cohort of apparently healthy individuals in Israel.”

We express our sincere appreciation to the editor and reviewers for providing valuable feedback that helped enhance the quality of our paper. Their constructive comments have enabled us to improve our manuscript significantly. We carefully considered each comment and provided a detailed response in the following pages. 

We hope the revisions will fulfill the requirements for publication in PLOS ONE. 

Thank you very much for your time and consideration. 

Sincerely, 

Michael Waisbourd, MD

Director, Glaucoma Research Center

Clinical Associate Professor of Ophthalmology Division of Ophthalmology

Tel-Aviv Medical Center

6 Weizmann Street Tel-Aviv, 64239, Israel Email: michaelwa@tlvmc.gov.il

Telephone: +972-3-6974165

Dear Reviewers,

Thank you for reviewing our manuscript. We truly appreciate your constructive comments and believe that your suggestions helped us improve our paper. 

Please see the responses below, highlighted on the revised version of the manuscript.

Journal Requirements

Reply: Done. 

Reply: Done. 

Reply: Done. 

4. In your Data Availability statement, you have not specified where the minimal data set underlying the results described in your manuscript can be found. PLOS defines a study's minimal data set as the underlying data used to reach the conclusions drawn in the manuscript and any additional data required to replicate the reported study findings in their entirety. All PLOS journals require that the minimal data set be made fully available. For more information about our data policy, please seehttp://journals.plos.org/plosone/s/data-availability.

Reply: Upon request, the authors agree to share data as part of a signed data sharing agreement, based on our hospital’s data sharing policy. 

Additional Editor Comments:

1. Though the sample size seems robust, it would be appreciated if the authors could comment on the power of study based on sample size and effect size.

Reply: 

We included sample size calculation in the revised version of the manuscript 

(Lines 108-110)

Reviewer 1

Overall impression:

In the manuscript entitled “The effect of body mass index reduction on intraocular pressure in a large prospective cohort of apparently healthy individuals in Israel“ it’s commendable that a large number of subjects were included as well a long period of examination, but the real goal of the examination remains unclear; is it: can we prevent glaucoma by losing body weight? Or?

Reply: 

Thank you for your comment. We cannot conclude from our study that weight loss can prevent glaucoma, however our results clearly show that BMI reduction in associated with IOP reduction. We can speculate that this IOP reduction may reduce the risk for developing glaucoma, however the study was not design to answer this question. Based on your valuable comment, we revised our conclusions accordingly: “The implications on prevention of glaucoma in healthy individuals and possible protective effect in patients with glaucoma require further investigation. 

(Lines 256-258)

Major comments:

1. The authors begin the introduction with the definition of glaucoma, but it is not clear

in the method section whether the subjects have glaucoma or some other eye disease.

Reply: 

Our study included participants that attended periodic health screening examination as part of the Tel-Aviv Medical Center’s Executive Health Program. We included in our analysis individuals that underwent IOP measurements, regardless of the status of other eye diseases. We added this limitation as possible confounder under study limitations 

(Lines 251-253)

2. In Methods is stated that ophthalmic evaluation included ocular history, visual

acuity and slit lamp; is it enough to confirm/rule out the existence of glaucoma or some other eye disease?

Reply:

Our comprehensive ophthalmic examination did identify eye diseases including glaucoma, however our analysis included all individuals who had IOP measurement, regardless of their final ocular diagnosis. We acknowledge this limitation in the revised manuscript as a possible confounding factor.

(Lines 251-253)

3. Methods: Which was inclusion/exclusion criteria for subjects who had IOP

measured?

Reply: 

We added these inclusion and exclusion criteria to the methods. We included in our analysis all individuals who had at least one IOP and BMI measurements at their baseline visit.

(Line 95-97)

4. The authors should explain why the subjects had the following examinations:

comprehensive laboratory and ancillary diagnostic tests, including blood chemistry

and metabolic profile, complete blood count, blood inflammatory markers, urine tests,

occult fecal blood test, prostate-specific antigen blood test (for men>40 years), cardiac

stress test, spirometry, audiometry, chest X-ray, and for women also gynecological

examination, including PAP test, physical breast examination, mammography, and

breast ultrasound.

Reply: 

All participants in Tel-Aviv Medical Center's Executive Health Program received these tests by default in order to assess their overall health and detect potential health disorders or diseases. Individuals from this cohort who consented to enroll in the Tel Aviv Medical Center Inflammatory Survey (TAMCIS) and signed an informed consent form were included in our study. This information is noted under Study Methods. Outcomes of some of the above-mentioned tests were published elsewhere (In the past 3 years: PMID: 29491490, PMID: 30647453, PMID: 30880949, PMID: 31175686, PMID: 31864299, PMID: 33426569). 

5. Methods: When was the second visit? What time period after the first visit?

Reply: 

The mean (SD) time difference between the first and second visit was 1.95 (0.9) years. (Line 183)

6. Results: Which was baseline IOP range?

Reply: The baseline IOP range was between 5 and 31 mmHg. We added this information to the revised manuscript. 

(Line 130)

7. Have any ophthalmological diseases appeared during the follow-up period?

Reply: 

As mentioned under reply to comment 1, other eye diseases were not evaluated in the current analysis, and this is now mentioned as a study limitation in our revised discussion (Lines 251-253)

8. Why are laboratory indices indicated in Table 1?

Reply: 

Since our study is addressing BMI and weight loss, especially among individuals with morbid obesity, we believe that adding baseline data on blood pressure and laboratory tests such as blood glucose level are relevant, since they may be directly related to obesity as part of the metabolic syndrome which includes obesity, blood pressure and diabetes. 

9. What is clinical practice?

Reply: 

We mention in our discussion: “Although our results were found in apparently healthy subjects, we can assume that BMI reduction will likely have a protective effect on patients with glaucoma, especially among morbidly obese individuals, since decrease of 1 mm Hg in IOP reduces the risk of glaucoma progression by 10%”. 

(Lines 224-227)

Reviewer 2

Overall impression:

The authors present an interesting study on the association between body mass index and intraocular pressure and how weight loss leads to a decrease in intraocular pressure. 

Reply: 

Thank you for your comment.

Major comments:

1. The study covers a large number of patients, appropriate for a study of this type in 

which patients have to be subdivided into different subgroups according to body mass index. However, the authors comment that the study exclusively includes apparently healthy individuals, but the methodology described does not seem to support this statement sufficiently. Pachymetry or examinations of the optic disc are not performed, so it cannot be ruled out that certain individuals have glaucoma (normotensive, or that by making only an IOP measurement, possible ocular hypertension has not been detected due to IOP fluctuations, thin pachymetries etc). 

Reply: 

Thank you for your valuable comment. Although participants did undergo a comprehensive eye examination, we did not exclude patients that were diagnosed with eye diseases (including glaucoma). This is an important limitation of our study, which is now mentioned under study limitations as a possible confounder.

(Lines 251-253)

2. Furthermore, it is not clear whether individuals with IOPs above normal after two

IOP measures were withdrawn from the study. 

Reply: 

Subjects with elevated IOP after 2 measurements were not excluded from the study cohort. We clarified this in our methods section: “IOP … of all eligible individuals was used for the current analysis”.

(Line 102)

3. It is commented that individuals who have weight loss have experienced a

decrease in IOP but it is not detailed whether this weight loss has been part of a standardized protocol to which all obese patients have been subjected, to carry out the descriptive study or has been an individual weight loss according to the criteria of each patient. 

Reply:

Our results are based on observation of obese individuals who lost weight without any

standardized protocol as part of the study. This information was added to the revised

manuscript 

(Line 94)

4. The discussion is a bit simple and poor, there are no comments on the glaucoma

association with all the blood parameters that have been determined and explained in the results and neither with the height of the patients and the smoking habit.

Reply:

We expanded the discussion based on your suggestions and now address association between glaucoma and parameters as well as smoking habits. 

(Lines 230-232). 

Lack of correlation between height and IOP is shown in Figure 3 (right panel) and is now also mentioned in the revised manuscript. 

(Line 180)

5. The bibliography is somewhat outdated; 

Reply:

The bibliography has been updated. 

We added the following references:

a. Law SM, Lu X, Yu F, Tseng V, Law SK, Coleman AL. Cigarette smoking and glaucoma in the United States population. Eye (Lond). 2018;32(4):716-25 

(Line 231)

b. Ng Yin Ling C, Lim SC, Jonas JB, Sabanayagam C. Obesity and risk of age-related eye diseases: a systematic review of prospective population-based studies. Int J Obes (Lond). 2021;45(9):1863-85 

(Line 199)

c. Panon N, Luangsawang K, Rugaber C, Tongchit T, Thongsepee N, Cheaha D, et al. Correlation between body mass index and ocular parameters. Clin Ophthalmol. 2019;13:763-9 

(Line 199)

d. Jabbehdari S, Chen JL, Vajaranant TS. Effect of dietary modification and antioxidant supplementation on intraocular pressure and open-angle glaucoma. Eur J Ophthalmol. 2021;31(4):1588-605 

(Line 216)

e. Reddy A, Halenda K, Cromer P, Chen L, Butler J, Raed A, et al. The Association of Intraocular Pressure With Obesity and Cardiometabolic Risk in a Young Farmworker Population. J Glaucoma. 2021;30(1):24-31. 

(Line 58)

6. The possible association between the measured parameters and IOP, and its possible clinical relevance, need to be further explained. 

Reply:

As suggested, we discuss in the revised manuscript measured parameters including blood glucose levels, smoking status and systemic blood pressure and their clinical relevance to glaucoma and intraocular pressure 

(Lines 237-242)

7. In the abstract, the conclusion part contains two sentences that are redundant since they say the same thing but with different words.

Reply:

The abstract conclusions were revised as suggested 

(Lines 41-43)

8. The beta coefficient of Table 3 is not explained in the table legend nor in the text; the meaning and interpretation of it should be addressed in order to better understand the results.

Reply:

We expanded on the statistical meaning of the beta coefficient in the caption of Table 3. In addition, odds ratio was calculated for each parameter. We also discuss these findings in more detail in the results section 

(Lines 164-168, Table 3)

---

## [Decision Letter · Decision Letter 1]

2 May 2023

The effect of body mass index reduction on intraocular pressure in a large prospective cohort of apparently healthy individuals in Israel

PONE-D-23-02619R1

Dear Dr. Savion Gaiger,

We’re pleased to inform you that your manuscript has been judged scientifically suitable for publication and will be formally accepted for publication once it meets all outstanding technical requirements.

Kind regards,

Asaf Achiron

Academic Editor

PLOS ONE

Additional Editor Comments (optional):

I believe this study can aid patients and doctors to understand better and control IOP in our community.

 Another topic I invite you to perform is studying the effect of sleeping hours on IOP. This can be done by measuring the hospital staff's IOP before and after a night shift. Such a correlation would be semifinal as well.

Well done for your corrections, and good luck.

Dr Asaf Achiron

Reviewers' comments:

Reviewer's Responses to Questions

**Comments to the Author**

1. If the authors have adequately addressed your comments raised in a previous round of review and you feel that this manuscript is now acceptable for publication, you may indicate that here to bypass the “Comments to the Author” section, enter your conflict of interest statement in the “Confidential to Editor” section, and submit your "Accept" recommendation.

Reviewer #1: All comments have been addressed

Reviewer #2: All comments have been addressed

2. Is the manuscript technically sound, and do the data support the conclusions?

Reviewer #1: Yes

Reviewer #2: Yes

3. Has the statistical analysis been performed appropriately and rigorously? 

Reviewer #1: Yes

Reviewer #2: Yes

4. Have the authors made all data underlying the findings in their manuscript fully available?

Reviewer #1: Yes

Reviewer #2: Yes

5. Is the manuscript presented in an intelligible fashion and written in standard English?

Reviewer #1: Yes

Reviewer #2: Yes

6. Review Comments to the Author

Reviewer #1: I have no comments for the authors of the manuscript entitled “The effect of body mass index reduction on intraocular pressure in a large prospective cohort of apparently healthy individuals in Israel“.

Reviewer #2: The manuscript has been improved and clarified by introducing the requested modifications. Now the objectives and conclusions of the study are more clearly explained and the references updated.

7. PLOS authors have the option to publish the peer review history of their article (what does this mean?). If published, this will include your full peer review and any attached files.

Reviewer #1: No

Reviewer #2: No

---

## [Editor Report · Acceptance letter]

9 May 2023

PONE-D-23-02619R1 

The effect of body mass index reduction on intraocular pressure in a large prospective cohort of apparently healthy individuals in Israel 

Dear Dr. Savion-Gaiger:

I'm pleased to inform you that your manuscript has been deemed suitable for publication in PLOS ONE. Congratulations! Your manuscript is now with our production department. 

Kind regards, 

on behalf of

Dr. Asaf Achiron 

Academic Editor

PLOS ONE